# The Treatment of Hepatocellular Carcinoma with Major Vascular Invasion

**DOI:** 10.3390/cancers16142534

**Published:** 2024-07-14

**Authors:** Tomoko Tadokoro, Joji Tani, Asahiro Morishita, Koji Fujita, Tsutomu Masaki, Hideki Kobara

**Affiliations:** Department of Gastroenterology and Neurology, Faculty of Medicine, Kagawa University, 1750-1 Ikenobe, Miki, Kita, Takamatsu 761-0793, Kagawa, Japan; tadokoro.tomoko@kagawa-u.ac.jp (T.T.); tani.joji@kagawa-u.ac.jp (J.T.); fujita.koji@kagawa-u.ac.jp (K.F.); tmasaki@saiseikai-kagawa.jp (T.M.); kobara.hideki@kagawa-u.ac.jp (H.K.)

**Keywords:** hepatocellular carcinoma, vascular invasion, portal vein tumor thrombus, bile duct tumor thrombus, hepatic vein tumor thrombus

## Abstract

**Simple Summary:**

Hepatocellular carcinoma with vascular invasion has a poor prognosis and inconsistent treatment evidence. Studies of portal vein tumor thrombus are relatively frequent; however, limited case studies of bile duct tumor thrombus and hepatic vein tumor thrombus exist. In this review, we also focus on bile duct tumor thrombus and hepatic vein tumor thrombus and review the current published studies.

**Abstract:**

Vascular invasion of hepatocellular carcinoma involves tumor plugs in the main trunk of the portal vein, bile ducts, and veins, and it indicates poor prognosis. It is often associated with portal hypertension, which requires evaluation and management. Treatment includes hepatic resection, systemic pharmacotherapy, hepatic arterial infusion chemotherapy, and radiation therapy. Recurrence rates post-hepatic resection are high, and systemic drug therapy often has limited therapeutic potential in patients with a poor hepatic reserve. Single therapies are generally inadequate, necessitating combining multiple therapies with adjuvant and systemic pharmacotherapy before and after hepatectomy. This narrative review will provide an overview of the treatment of hepatocellular carcinoma with vascular invasion.

## 1. Introduction

Primary liver cancer is the sixth most common cancer worldwide, with hepatocellular carcinoma (HCC) being the most prevalent, comprising 75–90% of all primary liver cancers [1,2,3]. Curative treatments include resection and radiofrequency ablation. In addition to curative treatments, such as resection and radiofrequency ablation, there have been remarkable advances in the treatment of intermediate- and advanced-stage HCC, including transcatheter arterial chemoembolization (TACE), hepatic arterial infusion chemotherapy (HAIC), molecular targeted agents (MTAs)/multi-target tyrosine kinase inhibitors (TKIs), and radiation therapy (RT). Despite these advances, HCC has a low 5-year survival rate because of a high recurrence rate after curative treatment, an increase in non-viral HCC such as fatty liver and alcohol, and the discovery of advanced HCC due to inaccurate surveillance [1,4].

Owing to its biological characteristics and liver anatomy, HCC often invades intrahepatic vessels, especially the portal venous system, and has a poor long-term prognosis [5]. HCC with vascular invasion is classified by the Japan Liver Cancer Association [6]. Among portal vein tumor thrombus (PVTT), Vp3 (thrombus tip in the ipsilateral first branch) and Vp4 (thrombus tip reaching the portal vein trunk or the more distal contralateral portal vein branch) are both considered clinically emergent and potentially fatal within 2 weeks [7]. In addition to the portal vein, HCC also causes tumor invasion of the bile ducts and hepatic veins, but case studies are limited.

In the Barcelona Clinic Liver Cancer (BCLC) staging system, vascular invasion is classified as an advanced stage, with sorafenib being the only recommended treatment [8], though its effectiveness is limited. Only a few patients with good liver function can undergo radical surgery. Therefore, in clinical practice, nonoperative treatments are often used, and combining these treatments is being explored to improve outcomes. This review discusses the current treatments for HCC with severe vascular invasion.

## 2. Materials

In this review, the authors conducted a comprehensive search for relevant studies using electronic databases including PubMed and MDPI from available peer-reviewed journals. The search was conducted using keywords related to HCC and vascular invasion, such as hepatocellular carcinoma, portal vein invasion, portal vein tumor thrombus, bile duct invasion, bile duct tumor thrombus, hepatic vein tumor thrombus, or inferior vena cava tumor thrombosis (n = 2157). Following the initial database search, the reference lists of identified articles were reviewed, and potentially eligible papers were selected for inclusion. Full-text access was obtained for the selected studies that met the inclusion criteria. Non-English articles or those deemed inappropriate were excluded. Quality assessment and data extraction were independently performed by two reviewers. Finally, 84 papers were extracted.

## 3. Portal Vein Tumor Thrombus

Owing to the biological characteristics of HCC and the liver’s anatomy, intrahepatic vessel invasion, especially PVTT, is the most common form of macrovascular invasion, occurring in 10% to 60% of patients with HCC [9,10]. Patients with PVTT have a median survival of 2.7 months without treatment [11], indicating a poor prognosis. PVTT involvement of the main portal vein causes portal hypertension, leading to gastrointestinal bleeding, ascites, and a decreased hepatic reserve. PVTT can also induce multiple intrahepatic tumor seeding and recurrence. In the BCLC staging system, PVTT is classified as advanced stage and sorafenib has been the sole recommended treatment option [8]. However, sorafenib offers only a modest survival benefit in patients with vascular invasion.

Thus, prior to 2020, sorafenib was the only effective first-line drug with evidence-based sequential treatment for advanced-stage BCLC. However, recent years have seen emerging evidence supporting various treatments. In the BCLC 2022 update [12], systematic treatment was recommended for advanced stages, reflecting this evolving landscape. However, the efficacy of drug therapy in patients with positive vascular invasion has only been reported as a subgroup analysis of phase III trials involving systemic drug therapies with MTA/TKI or immune checkpoint inhibitors (ICIs); trials specifically addressing highly vascular invasive cases are lacking. Reports suggest the potential utility of radiofrequency ablation therapy for PVTT extending into the main portal vein of HCC, though practical implementations remain limited [13]. Japanese guidelines for HCC [14] recommend hepatic resection for resectable cases and systemic therapy secondarily for unresectable cases, followed by poor recommendations for HAIC and TACE, with the evidence being weak. Furthermore, radiation therapy has become increasingly popular in recent years. Thus, we reviewed the research on advanced PVTT.

### 3.1. Hepatic Resection

Whether patients with HCC with PVTT should undergo hepatic resection is controversial owing to their higher metastatic risk, often leading to systemic therapy recommendations [2]. Historically, PVTT indicated an advanced stage and liver resection was not recommended. Surgical resection can be effective in patients with HCC whose PVTT is limited to the primary branches of the portal vein [15] and who achieve negative margin (R0) liver resection [10,15]. Median survival was 15 months in patients with large vessel invasion who underwent hepatic resection; 3-year survival was 33% and 5-year survival was 20% [16]. Japanese HCC practice guidelines [14] recommend hepatic resection for resectable cases. However, some believe that oncologic resectability should be considered in Vp3/4 cases of HCC with PVTT. Despite technically feasible resections, recurrence rates are high in these cases [17]. The superiority of surgical management over modern systemic treatment remains unclear and should be further investigated. Randomized controlled trials are needed to guide treatment allocation [18].

In the IMbrave050 trial evaluating postoperative pharmacotherapy, patients at a high risk of HCC recurrence after surgery received atezolizumab plus bevacizumab, which improved recurrence-free survival (RFS) compared to the follow-up group [19]. It is expected that preoperative and postoperative pharmacotherapy will appear in the future for high-risk groups, such as Vp3 or Vp4.

### 3.2. Systemic Therapy 

Systematic treatment is generally recommended for unresectable HCC [12]. If the patient has a good performance status and Child–Pugh classification A, the indication for combined immunotherapy should be considered. If these indications are absent, one of the TKIs should be selected. However, the therapeutic effect of systematic treatment in cases of severe vascular invasion (Vp3/4) is limited, and evidence supporting its efficacy remains scarce. The efficacy of systemic treatment with TKI or ICIs in patients with positive vascular invasion has only been reported as a subgroup analysis of phase III trials. Notably, the REFLECT and HIMALAYA trials excluded patients with HCC with Vp4 vascular invasion owing to their exclusion criteria Table 1.

In the SHARP trial, subgroup analysis demonstrated that sorafenib provided a benefit to patients with HCC with gross vascular invasion compared to the placebo, achieving a median overall survival (OS) of 8.1 months and a disease control rate (DCR) of 38.9% [20]. Subsequently, sorafenib became recognized as the only treatment to improve OS in unresectable HCC in randomized studies [26,27]. However, the SHARP study did not consider the extent of PVTT. In clinical practice, the efficacy of sorafenib for patients with PVTT is inadequate, with a median OS of 3.1 months and a median progression-free survival (PFS) of 2.0 months [28]. Consequently, there has been a pressing need for alternatives to sorafenib in this patient population.

HCC with Vp4 was an exclusion criterion in the REFLECT trial conducted to evaluate Lenvatinib. In clinical practice, lenvatinib has an overall response rate (ORR) of 53.8% and a DCR of 92.3% according to the Modified Response Evaluation Criteria in Solid Tumors (mRECIST) in patients with Vp3/4. These results are notably better than those observed with sorafenib [29]. However, although lenvatinib is safe and effective for advanced HCC in patients with a Child–Pugh A status [30], patients with a Child–Pugh B status were more likely to experience adverse events and may not respond adequately [31,32].

Cabozantinib is a multi-kinase inhibitor that blocks the VEGF receptor, MET, and AXL [22]. It is a second-line treatment for hepatocellular carcinoma after some form of systemic drug therapy. In the CELESTIAL trial, cabozantinib was more effective than placebo in HCC with macrovascular invasion, with a hazard ratio of 0.75 for OS and 0.42 for PFS [22].

The combination of atezolizumab and bevacizumab is currently the first-line treatment in advanced-stage (BCLC-C) HCC because it provides a superior survival benefit compared to sorafenib [12]. A subgroup analysis of OS in phase III trials showed that in the IMbrave150 trial, 48 patients with HCC with Vp4 received atezolizumab/bevacizumab combination therapy and reported a median OS of 7.6 months (hazard ratio, 0.62) and median PFS of 5.4 months (hazard ratio, 0.62) [23]. The Vp4 group was marginally inferior to the non-Vp4 group; however, it showed a high response rate not seen in previous TKIs [23,33]. Although the HIMALAYA trial of durvalumab–tremelimumab, which is discussed below, did not include patients with HCC with Vp4, atezolizumab and bevacizumab had a lower hazard ratio for OS and PFS than durvalumab–tremelimumabin comparison with sorafenib. Considering that the frequency of immune-related adverse events (irAEs) is also lower than that of durvalumab–tremelimumab, atezolizumab and bevacizumab are recommended as a first-line treatment for BCLC-C HCC owing to their wide benefits and reduced toxicity [34].

The STRIDE regimen (300 mg tremelimumab once and 1500 mg durvalumab every 4 weeks) significantly improves OS compared to sorafenib [25] and is indicated as a first-line therapy for adult patients with advanced or unresectable HCC [35].

However, HCC with Vp4 was not included in the HIMALAYA trial; thus, the benefit of durvalumab–tremelimumab therapy in highly vascular invasive HCC remains unclear. Durvalumab–tremelimumab may be the treatment of choice for patients with esophageal or gastric varices due to PVTT or in patients with impaired cardiac function who are not candidates for bevacizumab [35]. However, it should be noted that patients with gastric and esophageal varices with a risk of bleeding were also excluded from the HIMALAYA trial.

Overall, systemic treatment is recommended for unresectable cases of severe vascular invasion at this time. A major concern with introducing any systemic treatment is the coexistence of gastric or esophageal varices. In particular, bevacizumab carries a risk of rupturing gastric or esophageal varices; thus, patients with gastric or esophageal varices that are eligible for treatment should be treated first. In such cases, other systemic treatments, including the combination of tremelimumab and durvalumab, should be considered. In all clinical trials, including the HIMALAYA trial, patients with gastric and esophageal varices with a risk or bleeding were excluded. The bleeding risk of gastric and esophageal varices, the rate of tumor progression, and hepatic reserve should all be considered when determining treatment strategies.

### 3.3. Hepatic Arterial Infusion Chemotherapy

Patients with PVTT or a large intrahepatic tumor burden are less likely to achieve an objective response and are at a higher risk for hepatic deterioration after TACE; therefore, these patients are considered unsuitable for TACE [2]. Prior to the use of effective systemic treatments, HAIC was the treatment of choice for patients with large vessel invasion in Asian countries [36]. It was considered more effective than sorafenib, except in some cases where resection or TACE was indicated [33,37,38]. HAIC is still considered in cases where combined immunotherapy is not possible, such as in patients with gastric or esophageal varices or autoimmune diseases [39,40]. Even in patients with HCC with extrahepatic metastases, liver lesion progression is directly correlated with prognosis [41], making the control of intrahepatic lesions crucial [38].

Patients with vascular invasion have reduced hepatic reserve and are not amenable to systematic treatment, and alternative therapies such as HAIC are still often used in clinical practice. PFS of patients who received the atezolizumab and bevacizumab combination was significantly better than those who received HAIC (*p* < 0.05), but there was no significant difference in OS [42], with HAIC still being considered a useful treatment.

In Japan, hepatic arterial infusion reservoir therapy using a subcutaneous infusion port has been used for patients with severe vascular invasion. HAIC with 5-fluorouracil (5-FU) and cisplatin (FP) prolongs survival compared to symptomatic treatment with a median survival of 7.9 months for FP versus 3.1 months for symptomatic treatment [43]. A study comparing FP-HAIC and sorafenib also showed a median survival of 10.1 months for FP in patients with vascular invasion without extrahepatic lesions, compared with 9.1 months for sorafenib, indicating a prognostic benefit [44]. New-FP is a HAIC regimen consisting of powdered cisplatin and 5-FU suspended in lipiodol (oil-based contrast medium). The median OS for HAIC with New-FP is 16 months, and it is more effective than sorafenib [38,45,46]. HAIC-FOLFOX, which includes fluorouracil, leucovorin, and oxaliplatin, has been effective in locally advanced HCC, including Vp4; compared to sorafenib, it achieved a median OS of 13.9 months and tumor downstaging in 12.3% of patients [47]. However, the placement of the reservoir system requires skill and management of anticoagulation and infection, which may hinder its widespread adoption as a standard treatment.

HAIC is a catheter-based local treatment that differs from conventional systemic chemotherapy by delivering high concentrations of drugs locally with minimal systemic side effects. However, a limitation is the availability of facilities capable of performing HAIC, which restricts its widespread use.

### 3.4. Radiation Therapy

Historically, radiotherapy has been used in the treatment of HCC, primarily for palliative purposes owing to liver sensitivity to radiation, tumor depiction technique, and treatment technique certainty [48].

However, the advent of stereotactic body radiation therapy (SBRT) following particle therapy has established it as a viable treatment option for patients with localized disease for whom resection, transplantation, or radiofrequency ablation is not indicated. SBRT is not limited by tumor size or location and can be applied more broadly [49].

The median follow-up for patients with HCC with Vp3 or Vp4 who received radical proton therapy was 33.5 months, and the 5-year OS rate was 25.1% [50]. A meta-analysis comparing three-dimensional conformal radiation therapy (3D-CRT), selective internal radiation therapy, and SBRT for portal vein invasion cases indicated that SBRT demonstrated a significantly higher response rate compared to the other modalities. However, there was no significant difference in survival outcomes among the three treatment approaches [51].

Despite the increasing application of SBRT in HCC treatment, the optimal SBRT dose has not yet been determined [48].

Radiation therapy is a useful treatment option for unresectable HCC. Furthermore, its value in combination with systematic therapy and HAIC has also been reported, as discussed below. Recently, particle therapy has been an effective treatment option that can overcome the shortcomings of conventional radiotherapy; however, its major drawback is the limited availability of treatment facilities. As more treatment facilities become available, the clinical application of SBRT is expected to expand further.

### 3.5. Combination Therapy

As mentioned earlier, single therapy for HCC with PVTT has been reported to be insufficient. Therefore, combined therapy is often used for better treatment outcomes.

Patients at high risk of postoperative HCC recurrence, as described earlier, received atezolizumab plus bevacizumab and demonstrated improved RFS compared to the follow-up group [19].

In COSMIC-312, which compared the combination of cabozantinib and atezolizumab with sorafenib for advanced HCC, this combination was superior to sorafenib for OS and PFS in patients with hepatitis B virus infection and those with extrahepatic lesions or large vessel invasion at baseline [52].

HAIC is often combined with other therapies. Performing radiofrequency ablation prior to HAIC to shrink tumors in patients with HCC with PVTT can improve prognosis [53].

Combining 3D-CRT for PVTT in patients who do not respond to HAIC may result in prolonged OS [54]. Three-dimensional CRT radiotherapy targeted to localized areas of PVTT is more effective than sorafenib [46,55].

HAIC with radiation therapy for vascular invasion (HAIC + RT) resulted in an OS of 9.9 months, PFS of 3.9 months, and post-progression survival of 3.7 months, with the HAIC + RT group displaying significantly longer post-progression survival than the sorafenib group [56]. Furthermore, it is considered that HAIC + RT may lead to secondary treatment [46]. In advanced HCC with Vp4, the combined treatment of HAIC and RT was concluded to be effective, with an ORR of 13.7% for the primary tumor and 51.0% for Vp4, without the development of liver failure [57].

Subgroup analysis of the SILIUS trial in patients with HCC with Vp4 showed that patients treated with HAIC in combination with sorafenib had a better prognosis than patients treated with sorafenib alone [58]. In a phase III trial comparing sorafenib plus FOLFOX HAIC with sorafenib alone, OS and PFS were significantly longer with sorafenib plus FOLFOX HAIC than with sorafenib alone. HAIC-FOLFOX plus sorafenib is a multimodality treatment, and it is also considered a statistically superior option for unresectable HCC with PVTT [59].

Lenvatinib and HAIC, a cisplatin regimen, was well tolerated in patients with advanced HCC with an ORR of 64.7%, PFS of 6.3 months, and OS of 17.2 months using a mRECIST evaluation [60].

A phase II study of concurrent nivolumab and external beam radiation therapy (EBRT) in HCC with macrovascular invasion showed a median PFS of 5.6 months, a median OS of 15.2 months, and a median time to progression of 5.6 months, suggesting a high potential benefit [61].

Atezolizumab and bevacizumab combined with radiation therapy, particularly patients with Vp4 PVTT, bile duct involvement, or >50% hepatic involvement, has shown significant improvements in PFS and OS. In the high-risk population in this study, the median OS was 10 months and the median PFS was 6.50 months [62].

In patients with advanced HCC complicated by PVTT, the combination therapy of HAIC with camrelizumab (a selective humanized IgG4 monoclonal antibody) and rivoceranib (a highly selective vascular endothelial growth factor receptor-2 inhibitor) demonstrates improved survival outcomes and a better tolerated safety profile compared to the dual-drug combination of camrelizumab plus rivoceranib (mOS: 19.60 months vs. 11.50 months, *p* < 0.0001 and mPFS: 10.0 months vs. 5.6 months, *p* < 0.0001) [63].

Anlotinib, which is available in China, is a novel multi-targeted tyrosine kinase inhibitor that is administered as a basal therapy before TACE and RFA in patients with Vp2-4 HCC, with a median OS of 13 months and a 1-year OS rate of 64.3%. Its median time to tumor progression is 7 months and shows improved prognosis [64].

In patients with advanced HCC complicated by PVTT, combination therapy may provide better outcomes than single therapy. It is expected that more research will advocate appropriate treatment regimens.

## 4. Bile Duct Tumor Thrombus

HCC with bile duct tumor thrombus (BDTT) is a rare and unique entity of HCC in clinical practice compared to PVTT, with a prevalence of 1.2–12.9% [65]. BDTT may result in tumor cell infiltration within the bile duct, forming intrabiliary thrombi and obstructing bile flow [66]. Obstructive jaundice can lead to biliary infections and biliary hemorrhage, rapidly worsening the prognosis of patients with HCC. In patients with obstructive jaundice, successful biliary drainage is associated with improved OS [67]. Therapies such as radio ablation and TACE carry a considerable risk of biliary complications in patients with HCC with BDTT.

Some patients with obstructive jaundice can be treated with hepatic resection and have good long-term outcomes [67,68]. In a large retrospective study, patients with HCC with bile duct involvement had a median OS of 4.1 months. The surgery group achieved optimal survival with a median OS of 11.5 months, whereas other treatment modalities only achieved a median OS of ≤6 months [67]. A multicenter study of hepatic resection with preservation of the extrahepatic bile ducts showed excellent OS and RFS rates of 28.6 and 8.9 months, respectively [69]. Accurate diagnosis of the extent of hepatic resection is crucial, and cholangiopancreatic endoscopic procedures for diagnosing and managing jaundice play a valuable role in this regard [66]. However, patients with HCC with BDTT have a relatively high risk of early recurrence even after radical hepatectomy or liver transplantation [70]. Therefore, evidence for a useful treatment for recurrence after radical surgery is needed.

Evidence regarding the therapeutic efficacy of systemic chemotherapy for HCC with BDTT remains insufficient and is limited. Tanaka et al. reported that if obstructive jaundice was controlled by biliary endoscopy, HCC with BDTT could be treated with sorafenib and HCC without BDTT, with an OS of 14.1 months [71].

Lenvatinib has been found to be safe and effective for advanced HCC in patients with a Child–Pugh A status, including those with portal vein or bile duct involvement, despite being excluded from the REFLECT trial [30]. Lenvatinib treatment had an objective response rate of 85.7% and a DCR of 100% for patients with HCC with BDTT [30]. Alternatively, lenvatinib was more likely to cause adverse events in patients with a Child–Pugh B status and may not be fully effective [31,32]. Combination treatment with atezolizumab and bevacizumab was effective and well tolerated in the high-risk population, which included patients with bile duct involvement. However, significantly more high-grade adverse events and gastrointestinal bleeding were observed in the high-risk group than in the non-high-risk group [62]. In the IMbrave150 trial, patients with high-risk conditions such as tumor invasion of the portal vein main trunk, tumor occupancy ≥50%, and bile duct involvement were treated with atezolizumab and bevacizumab. OS was 7.6 months, and the objective response rate was 25.0% [24].

Few reports exist on the benefits of tremelimumab and durvalumab in HCC with BDTT, as BDTT was an exclusion criterion in the HIMALAYA trial. Only a few cases of HCC with BDTT developed irAE after 6 months of treatment with tremelimumab and durvalumab yet achieved a complete response. Further case studies are needed to understand this treatment approach [72].

Proton beam therapy has shown effectiveness and safety in patients with HCC and in patients with BDTT [73]. Studies have reported an OS of 19.9 months, a 1-year cumulative local recurrence rate of 5.3%, and a 1-year PFS rate of 58.9%, demonstrating promising results [74]. However, owing to the limited number of cases in these studies, future research is needed.

In a comparison of lenvatinib plus PD-1 versus HAIC with lenvatinib and PD-1 (HAIC-LEN-PD1) in patients with HCC with high-risk characteristics, the HAIC-LEN-PD1 group showed promising outcomes. They achieved an OS of 19.3 months, PFS of 9.6 months, a modified RECIST ORR of 76.7%, and a DCR of 92.2% based on modified RECIST criteria [75].

Because HCC with BDTT is a rare disease and evidence for treatment is lacking, more large multicenter studies are needed to provide supporting information.

## 5. Hepatic Vein Tumor Thrombus

Hepatic venous tumor thrombosis (HVTT) in patients with HCC is rare compared to PVTT, and little is known about it. The frequency of invasion into the inferior vena cava and right atrium is 1–4% [76]. Inferior vena cava tumor thrombus (IVC-TT) causes the tumor plug to extend into the inferior vena cava and right atrium, leading to complications such as secondary Budd–Chiari syndrome, pulmonary embolism, and pulmonary metastases [77]. This can cause sudden pulmonary embolism, refractory heart failure, and sudden death if the tumor plug spreads into the heart. The prognosis is poor, with a median survival time of 2–5 months without treatment [5,77]. With careful patient selection, aggressive treatment other than best supportive care may benefit patient survival [78]. Although sorafenib is the only recommended treatment for intrahepatic tumor plugs per BCLC guidelines, its effectiveness is limited, necessitating the development of new treatments.

Hepatic resection is associated with better prognosis in patients with HCC and HVTT, especially those without PVTT [79]. For HCC that has developed into IVC/right atrium, the mean survival is 19 months for the liver resection and thrombectomy group, 4.5 months for the TACE group, and 5 months for those receiving symptomatic treatment.

There is a small number of case reports on percutaneous ultrasound-guided radiofrequency ablation or percutaneous microwave ablation for treating patients with HCC and IVC-TT [80,81]. However, more studies of cases are in need to determine their efficacy for use in practical clinical applications.

The efficacy of systemic treatment with HVTT alone remains unknown as studies have primarily analyzed treatments for large vessel invasion, and the efficacy of HVTT alone is unknown. Median overall survival was 8.9 months compared to patients without macrovascular invasion [82]. PFS was 138 d in patients with Vv1-3 tumor thrombus, studied in combinations such as lenvatinib with PD-1 inhibitors (camrelizumab, scintilizumab, tisrelizumab) and HAIC [83].

Radiotherapy has shown utility in treating patients, with a meta-analysis of EBRT showing 1- and 2-year OS rates of 53.6% and 36.9%, respectively. Response and local control rates were 59.2% and 83.8%, respectively [84]. Radiation therapy with either 3D-CRT, intensity-modulated radiation therapy, or stereotactic body radiation therapy resulted in a median overall survival of 9.4 months and a 1-year OS rate of 37.1%. The ORR was 84.2%, with a local control rate at the last follow-up was 89.4% [85]. Pulmonary metastases prior to radiotherapy independently predicted OS in patients treated with radiation therapy for inferior vena cava tumor thrombi [86].

Combining TACE for HCC with IVC-TT and 3D-CRT for IVCTT showed a response rate of 71.4% and a median OS of 11.7 months, which showed to be more effective than TACE alone [87]. In a study of the efficacy of HAIC with 5-FU and systemic interferon (IFN)-α (HAIC-5-FU/IFN) for HCC with hepatic vein trunk (Vv2) or inferior vena cava (Vv3), the median survival time was 7.9 months. Fourteen of thirty-three patients were randomized to 3D- CRT to shorten tumor plugs in these patients, significantly improving survival compared to the non-combination group [88].

Combining radiotherapy with systemic systematic treatment shows promise. There is a case report of a complete response with a combination of sorafenib and radiotherapy, though more cases need to be accumulated for validation [89].

Combination therapy is important for hepatocellular carcinoma with major vascular invasion. The advantages and disadvantages of both conventional and modern treatment methods should be fully understood.

TACE, transcatheter arterial chemoembolization; HAIC, hepatic arterial infusion chemotherapy; MTA, molecular targeted agents; ICI, immune checkpoint inhibitors.

## 6. Conclusions and Future Directions

Advanced HCC is prone to vascular invasion, which considerably impacts prognosis. There are limited curative treatments for HCC with vascular invasion. Once successfully treated, these high-risk patients are prone to recurrence, and their liver function may decline during treatment. Reports of effective treatments for these highly advanced cases are scarce, and large-scale investigations are lacking owing to the small number of cases with severe vascular invasion. Current systemic therapies alone often provide insufficient responses, prompting efforts to improve outcomes by combining various therapies. Liver resection may improve prognosis, and the first step is to determine whether surgery is indicated. If surgery is not possible, systematic therapy, radiation therapy, or a combination of therapies can be utilized. HAIC can be considered for those who cannot be treated with these therapies (Figure 1, Table 2). In clinical practice, it is desirable to develop treatment strategies quickly while taking measures to prevent sudden death due to gastrointestinal bleeding, infection, liver failure, and heart failure. Developing a treatment strategy that combines multiple therapies at the right time is crucial, rather than relying on a single therapy for a patient.

## Figures and Tables

**Figure 1 cancers-16-02534-f001:**
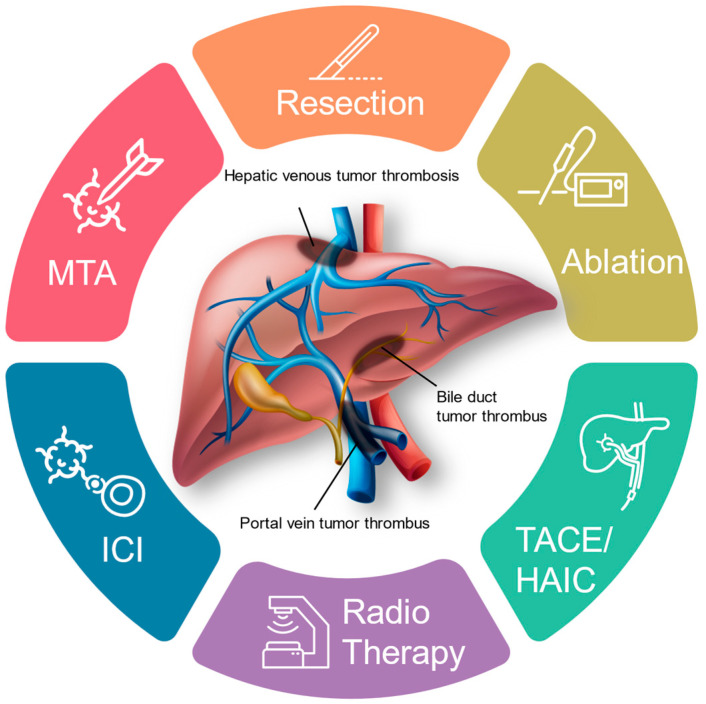
Treatment of hepatocellular carcinoma with major vascular invasion.

**Table 1 cancers-16-02534-t001:** Effects of systemic therapy for hepatocellular carcinoma with vascular invasion.

		Overall Population		Macrovascular Invasion		References
		Hazard Ratio	Median OS (Months)	Hazard Ratio	Median OS (Months)	
SHARP trial	sorafenib	0.69 (0.55–0.87)	10.7	0.68 (0.49–0.93)	8.1	[20]
	placebo		7.9		4.9	
REFLECT trial *	lenvatinib	0.92 (0.79–1.06)	13.6	0.87 (0.73–1.04)	11.5	[21]
	sorafenib		12·3		9.8	
CELESTIAL trial	cabozantinib	0.76 (0.63–0.92)	10.2	0.75 (0.54–1.03)	-	[22]
	placebo		8.0		-	
IMbrave150 trial	atezolizumab/ bevacizumab	0.66 (0.52–0.85)	19.2	0.68 (0.47–0.98)	7.6	[23,24]
	sorafenib		13.4		5.5	
HIMALAYA trial *	durvalumab/ tremelimumab	0.78 (0.65–0.93)	16.4	0.78 (0.57–1.07)	-	[25]
	sorafenib		13.8		-	

* Exclude cases of portal vein trunk invasion. OS, overall survival; -, no data.

**Table 2 cancers-16-02534-t002:** Recommended therapy for hepatocellular carcinoma with major vascular invasion.

Type of Invasion	Treatment Precautions	Recommended Therapy	References
portal vein tumor thrombus	portal hypertension	hepatic resection	[15,16,17,67,68,69,79]
	varices		
	ascites	systemic therapy	[20,21,22,23,24,25,30,31,32,62,71,72,82,83]
	liver failure		
bile duct tumor thrombus	obstructive jaundice	hepatic arterial infusion chemotherapy	[33,36,37,38,39,40,42,45,47]
	biliary hemorrhage		
	liver failure	radiation therapy	[50,51,73,74,84,85,86]
hepatic vein tumor thrombus	Budd–Chiari syndrome		
	pulmonary embolism	combination therapy	[46,52,53,54,55,56,57,58,59,60,61,62,63,64,75,87,88,89]
	heart failure		

## Data Availability

Not applicable.

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
