# Peer review of "The Treatment of Hepatocellular Carcinoma with Major Vascular Invasion"

_cancers, 2024, doi:10.3390/cancers16142534_

Round 1

Reviewer 1 Report

Comments and Suggestions for Authors

Even not new, the manuscript covers an interesting topic and it`s well written I have only minor comments for the Authors

- have you got some prevalence/incidence/mortality estimates about HCC with bile duct tumor thrombus or hepatic vein tumor thrombus? please, report them

- can we consider your paper either a systematic or narrative review? please, specify it just in title/abstract

- since you selected 84 papers, we would love to look at a general summary table, about their main characteristics

 - table 1, are we dealing with univariate or multivariate models too?

- have you got any survival data on PFS too? if any, report it

Author Response

I marked with blue marker the statement regarding the response to reviewer 1.

1,- have you got some prevalence/incidence/mortality estimates about HCC with bile duct tumor thrombus or hepatic vein tumor thrombus? please, report them

Thank you for your important remarks. We have added the description.

2,- can we consider your paper either a systematic or narrative review? please, specify it just in title/abstract

Thank you for your important remarks. This paper is a narrative review. I have added it to the abstract section.

 3,- since you selected 84 papers, we would love to look at a general summary table, about their main characteristics

Thank you for your important advice. I have followed your advice and prepared Table 2 as a summary of the literature.

4, - table 1, are we dealing with univariate or multivariate models too?

Thank you for your important remarks. Table 1 lists the hazard ratio and median OS for each trial, and no analysis was performed to compare them.

5,- have you got any survival data on PFS too? if any, report it

Thank you for your important remarks. I have added the description to the text according to your comment. Several references did not mention about PFS.

Reviewer 2 Report

Comments and Suggestions for Authors

The topic of this review manuscript is of interest; however, several deficits need to be improved for the acceptance of Cancers.

1. There are many typos and grammatical errors in the current form of this review manuscript that need to be carefully revised.

2. For the convenience of the readers, more representative figures for the text of this manuscript are highly recommended for this manuscript.

3. For important scientific descriptions, the reference citation should be more extensive, not just citing 1 or 2 papers.

4. It would be nice to have a more in-depth discussion about this topic of this manuscript.

Comments on the Quality of English Language

Moderate editing of English language required

Author Response

I marked with yellow marker the statement regarding the response to reviewer 2.

1. There are many typos and grammatical errors in the current form of this review manuscript that need to be carefully revised.

Thank you for pointing this out. This paper has been edited by a native English speaker. I have carefully checked and corrected the grammar and other details.

2. For the convenience of the readers, more representative figures for the text of this manuscript are highly recommended for this manuscript.

Thank you for your important comments. I have changed the figure and figure regend to be more detailed.

3. For important scientific descriptions, the reference citation should be more extensive, not just citing 1 or 2 papers.

Thanks for your important comments. I have followed your advice and added some references.

 4. It would be nice to have a more in-depth discussion about this topic of this manuscript.

Thank you for your comment. I have taken your advice and added my views on treatment to the Conclusion and Future Directions part.

Round 2

Reviewer 2 Report

Comments and Suggestions for Authors

The authors have addressed the reviewer'scomments.